# Diagnostic Methods Used in Detecting Syphilis in Paleopathological Research—A Literature Review

**DOI:** 10.3390/diagnostics15091116

**Published:** 2025-04-28

**Authors:** Grzegorz Mikita, Michalina Jagoda Lizoń, Julia Gąsiorowska, Maciej Mateusz Hanypsiak, Jan Falana, Mateusz Mazurek, Oliwier Wojciech Pioterek, Krzysztof Wolak, Joanna Grzelak, Dominika Domagała, Dariusz Nowakowski, Paweł Dąbrowski

**Affiliations:** 1Vertex—Paleoanatomy Students Scientific Club, Wroclaw Medical University, 50368 Wroclaw, Poland; michalina.lizon@student.umw.edu.pl (M.J.L.); julia.gasiorowska@student.umw.edu.pl (J.G.); maciej.hanypsiak@student.umw.edu.pl (M.M.H.); jan.falana@student.umw.edu.pl (J.F.); 2Clinical and Dissecting Anatomy Students Scientific Club, Wroclaw Medical University, 50368 Wroclaw, Poland; mateusz.mazurek@student.umw.edu.pl (M.M.); oliwier.pioterek@student.umw.edu.pl (O.W.P.); krzysztof.wolak@student.umw.edu.pl (K.W.); 3Student Scientific Association at Department of Physical Education and Sport, Wroclaw Medical University, 50368 Wroclaw, Poland; 4Division of Anatomy, Department of Human Morphology and Embryology, Wroclaw Medical University, 50368 Wroclaw, Poland; joanna.grzelak@umw.edu.pl (J.G.); dominika.domagala@umw.edu.pl (D.D.); pawel.dabrowski@umw.edu.pl (P.D.); 5Department of Anthropology, Wrocław University of Environmental and Life Sciences, 51630 Wroclaw, Poland; dariusz.nowakowski@upwr.edu.pl

**Keywords:** syphilis, anthropologic diagnostics, long bones, paleopathological analysis, anthropological methods

## Abstract

Syphilis is a disease caused by Treponema pallidum. It is primarily transmitted sexually or vertically during pregnancy. The origin is twofold, namely, it comes from America or Europe. Syphilis was first recorded in a human skeleton in the 11th century. However, signs of treponemal disease were observed in osteological material from a Pleistocene bear. Hence, it is necessary to study syphilis on bone material to better understand the etiology of the above disease and, consequently, introduce preventive measures. Examination of syphilis on skeletal material can be performed at the macroscopic and microscopic levels. Those methods refer to the visual assessment of skeletal material, namely the identification of characteristic pathological changes caused by syphilis, such as periostitis, which manifests itself as thickenings on the bone surface, and syphilis nodules (gummata), which are defects in the bones. Most often, these changes are found on long bones such as the tibia, femur, and skull. Radiological methods may be used, such as X-ray, computed tomography (CT), Micro-CT (ICT), as well as molecular examination. Summarizing, this review is an overview of the latest methodology regarding syphilis research on skeletal material, thanks to which it can better understand its genesis.

## 1. Introduction

Paleopathology, studying ancient diseases through remains, reconstructs past health patterns. It identifies diseases, such as syphilis, Hansen’s disease (leprosy), and tuberculosis, via analyzing osteological remains [1]. The genesis of syphilis remains unclear. The most important hypotheses are that it developed in America and came to the Old World with the Columbus expedition or was present on the European continent before [2]. Syphilis played a great role in mankind’s history and persists as a threat to human health. The global incidence of syphilis in 2019 was over 14,000,000. Consequently, paleopathological research on syphilis plays an important role in better understanding the etiology of the above disease and, therefore, the possibility of faster implementation of preventive measures [3]. It is essential to consider the natural progression of the disease. Following infection, syphilis enters an incubation period lasting several weeks to months, after which the primary stage emerges. This phase typically lasts for a few weeks and is followed by the secondary stage. The disease then enters a prolonged latent phase, which may persist for years before progressing to the tertiary stage —at which point osseous manifestations may develop [4,5,6,7,8]. The skeletal elements most affected in venereal syphilis are the tibia and the cranium [9,10]. It is estimated that bone lesions are present in approximately 10% to 20% of skeletal remains of individuals who suffered from venereal syphilis [5]. These cases are typically associated with the tertiary stage of the disease [8]. However, it is important to note that the incidence of bone involvement is significantly higher in congenital syphilis, ranging from 10% to 65% for dental changes and even up to 86% for clavicle lesions [11,12].

Nowadays the methodology for syphilis on bone material includes macroscopic analysis, namely relying on visual examination such as periosteal reactions and caries sicca, which are characteristic of tertiary syphilis, for instance, by X-ray, Computed tomography (CT), or Micro-CT (ICT) [13] Another method is microscopic or histological analysis, which assures the evaluation of bone remodeling patterns. Molecular techniques rely on polymerase chain reaction (PCR) and ancient DNA (aDNA) analysis, showing direct evidence of Treponema pallidum infection [14]. Recently, it has been postulated that shotgun metagenomics may be another method for detecting syphilis on skeletal materials. This technique focuses on sequencing the genomes of entire microbial communities, without the need to culture them first [15].

Precious information about the past and present of this disease can be obtained in paleopathological studies. They require diagnostic techniques, which allow distinguishing its marks in human remains. We present a literature review of such methods described in the literature until now. Our objective was to describe techniques used in diagnosing syphilis in paleopathological samples, especially in osseous remains, along with an evaluation of the strengths and limitations of each approach. We hope that this review, by consolidating current knowledge on the diagnostic methods used in detecting syphilis in paleopathological material, will facilitate future research efforts and serve as a valuable resource for other investigators.

## 2. Materials and Methods

As there is a lack of comprehensive reviews specifically on methods of diagnosing syphilis in bone material, the aim of this study was to cover as many publications as possible from as wide a time frame as possible. Therefore, articles from 1975 to 2024 were included in the review, as well as one from 1955. This enabled us not only to examine the techniques themselves but also to trace their development, changes, and improvements. A literature review was conducted in accordance with a predefined methodological protocol:

We developed the following search terms: syphilis AND molecular AND anthropology.syphilis AND markers AND anthropology.syphilis AND imaging AND anthropology.syphilis AND diagnostics AND anthropology.syphilis AND differentiation AND anthropology.

PubMed (National Library of Medicine, Bethesda, MD, USA), Embase (Elsevier, Amsterdam, The Netherlands), Web of Science (Clarivate, Philadelphia, PA, USA and London, UK), and Google Scholar (Google LLC, Mountain View, CA, USA) were browsed using them. Articles were first sorted by relevance, and we screened 100 of each search term in each base. This was performed between 16 July 2024 and 27 July 2024. Then, we reviewed the articles a second time, using the following criteria: The record was an original, peer-reviewed, and published study.The full text was available.The full text was in English.The full text was relevant to the topic of our review: it discusses how the described diagnostic methods are being or can be used to detect osteological lesions or molecular marks caused by *Treponema pallidum* infection in anthropological samples and/or possibilities of differential diagnosis of those lesions.

After the removal of duplicates, there were 124 papers. Then, another group of authors reviewed the articles. We included 91 of them, which fulfilled the criteria described above. Therefore, every included article was reviewed twice, after the screening phase, by two independently working researchers.

From each selected article, the following data were systematically extracted:The total number of diagnostic techniques employed in the study.The specific types of diagnostic techniques utilized.The chronological age of the examined samples.The kind of examined material.

Subsequently, the articles were classified into thematic groups based on the diagnostic method applied. In cases where multiple methods were implemented within a single study, the article was assigned to all relevant categories. Each assigned group was then subjected to independent analysis in terms of the respective diagnostic technique. The results of the review are presented in Figure 1.

## 3. Results

### 3.1. Characteristics of Included Studies

Characteristics of included studies are presented in Table 1.

### 3.2. Macroscopic Analysis of Bone Lesions

Upon reviewing the selected literature, it becomes evident that macroscopic analysis, also referred to as gross examination, plays a pivotal role in the detection of syphilis in paleopathological research. This method serves as a fundamental approach, enabling researchers to identify skeletal remains likely afflicted by syphilis and to exclude those lacking bone lesions pathognomonic of the disease [1,2,3,48,63,72]. 

In some studies, stereo microscopes or hand lenses are employed to observe the lesions at low magnification [20,48,65,73]. Additionally, researchers often weigh and measure skeletal material before detailed gross examination to facilitate comparison with unaffected specimens, thereby aiding in diagnosis [1]. Bony alterations, such as pits, cavities, and thickenings, are also measured to provide a more detailed description of the material [20,24,83]. Researchers examine para-articular and articular bony surfaces under raw operating light [73,82] and describe the type and distribution of lesions to conduct accurate diagnoses [63]. 

Syphilis in paleopathological material is distinguished not only by alterations in bone structure but also by their pattern of occurrence and the specific types of bones affected [50]. Some authors emphasize the significance of reassociating skeletal elements derived from a single individual that became disarticulated post-mortem, as this process aids in understanding lesion distribution in the afflicted remains [50]. 

Fragmented bones may undergo a process of restoration, enabling the observation of complete or nearly complete specimens [57]. The long bones, such as the tibia, femur, humerus, ulna, and radius, along with the cranium, are the most frequently affected, with the tibia being particularly prone to alteration [3,4,9,31,75]. Indeed, tibial lesions are present in 99% of individuals suffering from treponemal disease [65]. Alterations may also be observed in the clavicles, vertebrae, scapulae, ribs, and sternum [4,20]. Destruction of the frontal bone is considered characteristic of venereal syphilis, as is the involvement of the forearms and the tibia [10,25,94]. 

Given that syphilis is a systemic disease, the lesions are diffuse and broadly distributed across the skeleton [89]. They are present in 10–20% of individuals [4] and can be categorized as either unremodeled (active) or remodeled lesions [5]. Sclerotic, fibrous periosteal reaction with pitting is an active lesion highly suggestive of venereal syphilis [5]. Hypertrophic periostitis causes fusiform expansion and thickening of the shaft of long bones caused by the formation of subperiosteal new, woven bone over the normal, smooth cortex [5]. Sometimes, new bone appears in the form of raised plaques and spicules, which fuse and create bridges across shallow vascular grooves [51]. The surface of the new bone displays coarse striations and is porous, rugose, containing numerous pits (enlarged vascular foramina), as well as larger depressions with sclerotic bone at their borders (Figure 2A) [1,72,77,79]. 

In severe cases, the pathological process obliterates physiological bone surface markings [95], while the medullary cavity becomes progressively occluded by bone migration, with cancellous bone transforming into lamellar bone (Figure 2A) [6,9,25]. The latter can only be observed macroscopically at post-depositional breaks, e.g., in the broken-off shaft, illustrating the limitations of gross examination [5,6,51]. Excessive bone formation is also observed at muscle attachment sites [45,77,79]. 

The foregoing description encapsulates the diagnostic non-gummatous lesions of syphilis [4,10]. Saber shin tibiae and hints of gummatous lesions (reflecting the mixture of bone formation and destruction) are pathognomonic of treponemal infection but are rarely encountered in skeletal remains due to their late onset [4,31]. This also applies to long bone lesions, which primarily affect the diaphyses and do not typically exhibit bilateral characteristics until the disease reaches its tertiary stage [4,6,78,89]. 

Crania affected by syphilis often display a more lytic nature of lesions [45]. Oval or circular shallow depressions of the outer table of the cranium, most often located on the frontal bone, are characteristic of the disease [51]. These cavities tend to cluster and fuse, forming large, eroded areas [19]. The resulting lytic lesions, combined with irregular, sclerotic bone formed during the healing process, give rise to the characteristic caries sicca of the frontal bone, a hallmark of venereal syphilis (Figure 2B) [77]. This alteration appears as thick nodules separated by deep, stellate depressions, representing a gummatous lesion with a ‘worm-eaten’ appearance [9,33,53,58,79,94]. 

Remodeled lesions, in contrast, constitute a healing process, resulting in the deposition of mature, smooth bone [5]. The co-existence of these alterations creates an irregular bone surface, giving the shaft an ‘inflated’ and ‘wrinkled’ appearance (Figure 2A) [77,79].

Ernest G. Walker, in his 1983 study, presented an interesting case of syphilitic aortic aneurysm, which manifested as erosion of the sternal end of the right clavicle and the left side of thoracic vertebral bodies, both caused by the pulsatile activity of an aneurysm [87]. Certain authors reported syphilis affecting joints, leading to the destruction of their surface [4]. 

Furthermore, macroscopic examination aids in differentiating between venereal syphilis and other treponematoses (congenital syphilis, yaws, and bejel, also known as endemic syphilis). While bone lesions occurring in the tertiary stage of venereal syphilis, yaws, and bejel are similar, subtle distinctions exist [62,63]. These differences are most noticeable on the scale of a population and thus are labeled population variations [51,58,63,65]. Here, we aim to underscore the critical importance of geographical, environmental, epidemiological, and historical contexts, which provide indispensable assistance in differentiating between the treponematoses [5,48,51,63,77,79]. 

Regarding congenital syphilis, dental characteristics play a pivotal role in achieving a definitive diagnosis. Hutchinson’s incisors and Moon’s molars are considered the only dental alterations pathognomonic of the disease [35,95]. Thus, reduced dimensions combined with thinned enamel and occlusal notching of upper permanent incisors make the diagnosis unequivocal [4,40]. Hutchinson himself described them as ’screwdriver’ shaped due to their constricted occlusal surface (Figure 2E) [96]. It should be noted that their characteristic vertical notch is often worn away during life and thus is less visible in skeletal remains. Conversely, in some cases, the notch has not yet formed, and a thin enamel patch occupies its place [35,55,95]. Certain sources also identify a depression on the labial surface of the incisors as characteristic of the disease [35,55]. Apical hypoplasia and notching of the permanent canines are less indicative of the condition and warrant careful consideration [40]. 

Given inconsistencies in the literature, we would like to clarify the distinction between Moon’s first molars, also referred to as bud molars, and the mulberry molars described by Fournier [35,55,65]. Molars described by Moon are dome-shaped with a narrow occlusal surface and crowded cusps clustered at the center, as presented in Figure 2C [40,97]. In contrast, Fournier’s first molars exhibit a heavily atrophied cusp base and sinuses penetrating the dentine (Figure 2D) [40,55,95]. In both instances, the teeth are reduced in size compared to adjacent molars and show enamel hypoplasia [36,40]. These pathologies can be observed on permanent teeth, while deciduous teeth are usually unaffected and differentiate congenital syphilis from yaws [78].

Syphilitic osteochondritis, manifesting as epiphyseal and metaphyseal erosion combined with cortical detachment and paradiaphyseal calcification, is characteristic of congenital syphilis, though it is less common than periostitis [6,48,82]. Periostitis, present in congenital and venereal syphilis as well as in yaws, complicates the differential diagnosis [6,66]. The well-known saber shin deformity, also termed ’pseudo-bowing’, caused by periostitis and subsequent subperiosteal bone formation, occurs more frequently in late congenital syphilis than in venereal syphilis [1], albeit it is also present in yaws and bejel in a less severe form [45,63]. However, only in congenital syphilis does saber shin remodeling progress to the extent capable of obliterating surface striating and rendering the diagnosis by visual examination almost impossible [66]. True bowing of the tibia is exclusive to childhood forms of treponematoses when the disease impacts immature bone, allowing anterior curving of the interosseous crest (Figure 3) [51,55]. 

As noted previously, the healing process in venereal syphilis leads to the deposition of new, smooth bone, occasionally obscuring all diagnostic markers visible to the naked eye [65]. By contrast, reactive bone remodeling in yaws produces a rugose surface, facilitating differentiation between these treponematoses [3]. Higoumenakis describes enlargement of the sternal end of the clavicle as highly indicative of late congenital syphilis, which may assist in making a correct diagnosis when the cranium with teeth is not available for examination [11]. 

The number of bones affected and the localization of lesions are instrumental in distinguishing between the treponematoses (Figure 4) [66]. For instance, tibial lesions in venereal syphilis are often unilateral, contrasting with the bilateral presentation in yaws and bejel [65]. Yaws tends to affect a greater number of bones on average [66]. Additionally, hand and foot lesions are characteristic of yaws, while the fibula is most often affected in bejel [63]. Nonetheless, a proper diagnosis cannot be conducted without defining the disease’s pattern and prevalence within the examined population and comparing it to populations diagnosed during life, ideally focusing on tibial lesions [11,65,66,67,84]. 

It is essential to examine the skeletal material for additional pathologies and exclude other potential diseases before making a definitive diagnosis. Macroscopic analysis alone may be insufficient for conducting a differential diagnosis and may require supplementary diagnostic methods [4]. The diffuse nature of syphilitic lesions, coupled with the productive rather than destructive character of bony alterations, distinguishes the disease from pyogenic osteomyelitis [1], which is often noted as highly resembling treponemal infection [77,79]. Moreover, syphilitic lesions usually lack the sequestration diagnostic of pyogenic osteomyelitis [77]. 

Active cranial lesions are similar to those seen in fungal infections such as cryptococcosis and sporotrichosis, but these primarily affect the vertebrae, ribs, hands, feet, and joints [51,77,78]. Altered teeth seen in congenital syphilis may be difficult to distinguish from so-called mercurial teeth, which occur in patients treated with mercury. Both display a similar atrophied appearance, with enamel loss and dentine exposure [37,60,93]. Notwithstanding their resemblance, mercurial teeth are suggestive of syphilis themselves since mercury was extensively administered to syphilitic patients [36,37,93]. Teeth presenting both lesions characteristic of congenital syphilis and mercurial treatment are referred to as syphilitic-mercurial teeth [36]. Mercury exposure may also cause erosion of the dental alveoli [38,93]. Before conducting a definitive diagnosis, it is crucial to exclude the possibility of mistaking antemortem lesions for taphonomic processes or postmortem damage. Researchers confirm this, for instance, through the examination of specimens at low magnification (c. x40) [65]. This is particularly applicable to osteolytic lesions with destructive character [6], as postmortem erosion resembles crumbling of the external cortex [53]. Only in postmortem erosion do deeper structures of the bone come into sight [53]. 

A similar approach applies to metastasizing neoplasms, which usually produce lesions penetrating beneath the bone’s outer layer [91]. Lytic lesions of syphilis also resemble those present in tuberculosis (TB). However, TB can be excluded if the subject of concern is the cranium, as it is rarely affected by this disease [24]. Lesions of the ribs are highly suggestive of tuberculosis and should be considered in differential diagnosis [41]. Moreover, distinguishing between bone fractures and old, remodeled lesions presents challenges, albeit it is worth noting that fractures do not exhibit the stellate shape, in the form of which syphilitic lesions sometimes occur [10,58]. Other microorganisms capable of passing the placental barrier must also be considered in the differential diagnosis of congenital syphilis, though only a few cause osteological symptoms [6]. 

It is noteworthy that researchers utilize macroscopic analysis not only as a preliminary screening tool but, in some cases, as a standalone method for definitive diagnosis [72]. The reliance on gross analysis alone varies depending on the study. However, it is crucial to recognize that gross examination is subject to potential biases and should be carefully considered before being used as the sole diagnostic approach for syphilis [9]. 

There are several limitations associated with diagnosis based on macroscopic examination of syphilitic bone lesions, the primary one being the condition of the skeletal remains themselves. Ideally, the specimens should be complete and well-preserved [19], but in reality, not all bones are in a sufficient condition for a proper examination [9,68,71,78,83]. 

In the matter of teeth, postmortem exfoliation of the enamel precludes the evaluation of their true width [40]. Additionally, visible bone lesions appear only in individuals who survived long enough for the disease to progress to its tertiary stage, where it can manifest in this specific manner [7]. As a result, skeletal lesions are relatively rare in syphilis [12,25,38]. 

Furthermore, some lesions characteristic of the disease are not visible with the naked eye, such as Wimberger’s sign of congenital syphilis [5,84], and in certain cases, pathognomonic lesions may be completely absent, necessitating the use of alternative diagnostic methods [9,17,68,71]. 

### 3.3. Microscopic Analysis of Bone Lesions

The microscopic examination of bone lesions proves to play a pivotal role in the detection and differentiation of syphilis signs in paleopathological research. It complements the macroscopic identification of the pathology, and by broadening the methodological approach, it potentially provides better diagnoses and assesses the preservation of bone and its constituent parts [45,65,86]. 

Histological thin sections are taken from visibly apparent pathological areas of the examined skeleton, and if possible, they are later compared with previously excavated and analyzed remains from the researched areas, thus resulting in finding common features and signs [24,68,86]. 

A microscopic examination aims at finding histological appearances that point to chronic, episodic, osteoclastic, and osteoblastic processes characteristic of syphilis [24,68,82]. It can use histological scales to make it more comprehensible. For example, one of these can be a histological index designed by Millard (2001) that can be used to make a general assessment of bone quality [86]. This index ranks histological bone sections from 0 to 5, with the bottom values (i.e., 0) signifying no original features identifiable except possibly Haversian systems, to the top values (i.e., 5), where structures are very well-preserved and virtually indistinguishable from modern bone. Over the years, there were set criteria for differentiating and confirming the syphilitic origin of the apparent bone lesions [45]. One of those criteria can be those described by Schultz (1994, 2001, 2003) [20,65,86]. 

Worth noting is that microscopic examination of syphilitic bones may reveal extensive osteolysis, which was then followed by bone regeneration [24]. The especially vulnerable region is the periosteal cell directly adjacent to the skin. Moreover, the suppressed osteosynthesis on the endosteal side can be caused by treatment using arsenic due to misdiagnosis. 

In the histological section, the mosaic structure of the diploe that is typical for this disease can be seen [65]. Numerous broad tangential lamellae can be found, which are partly interrupted by well-developed, relatively large blood vessel sinuses [68].

Another noticeable observation can be the presence of a Maltese cross in the polarized light, as it denotes good preservation of lamellae in osteon structures [64,82]. Subsequently, the diagnostic criteria described by Schultz are used for differentiating venereal syphilis from other specific and nonspecific inflammatory diseases. It states that the periosteal thickening of long bone shafts and the concomitant microscopic changes seen in venereal syphilis are a regular occurrence, most notable when the process of healing and remodeling of compact bone has not yet finished [86]. 

The primary criterion is the presence of lines or band-like structures called the “Grenzstreifen”, which separate the primary cortical bone from the active periosteal new bone layer laid down during bouts of infection. The next criterion is called the “Polster”, which represents a pillow-like villous proliferation of very dense parallel lamellae found in the highly thickened periosteal layer of cortical bone in the shaft of a long bone, e.g., the tibia. There can also be present sinuous resorption lacunae between the original bone surface and the newly calcified layers, which can suggest syphilis but are also found in other nonspecific inflammatory diseases [49,86]. 

The histological examination may not only show signs of existing pathology, but it can also indicate the age at which the first lesions formed and how long the process has been undergoing. Histological assessment can be based on the number of present fragments of remodeled osteons and the amount of non-osteal vascularization [86,91]. It is worth noting that the detection of interstitial lamellae, being the remains of earlier generations of lamellae postponed from the periosteal and endosteal side, may confirm the age estimated by the macroscopic analysis [24]. 

It is important to be aware of the possibility that besides the presence of microscopic signs in the bone section, there can be changes that are not of syphilitic etiology, such as microorganism focal destruction in the form of Wedl canals (described by Hackett, 1981), fungal and microorganism intrusion of unknown origin and, additionally, foreign materials, e.g., soil, crystals within lacunae, and/or Haversian systems can be visible [86]. Therefore, the lack of good osseous preservation may lead to some difficulties in discerning microscopic structures indicative of syphilis and, in some cases, even make it unattainable by the sole microscopic analysis, which indicates the importance of extending the methodology by other diagnostic means, which allows for an improved diagnosis and better understanding of the disease process Multiple lines of evidence in support of a hypothesis are fundamental in paleopathology [45,65,86]. 

Some studies use fluorescent microscopy besides common light microscopy. The use of fluorescence allows us to make visible and measurable compounds in the bone. As an example, it can reveal bone matrix saturation with hydroxyapatite and present inclusions of foreign compounds. Unfortunately, the analysis and interpretation can be disturbed by the presence of minor impurities [24,45,65,86]. 

### 3.4. Radiological Examinations

Radiological examinations represent the optimal solution and support for macroscopic examinations. They facilitate enhanced visualization of pathological changes in bones, facilitate differential diagnosis of bone lesions, and enable the exclusion of other bone-related diseases

#### 3.4.1. X-Ray

X-rays are a valuable tool in the diagnosis of infectious diseases such as whooping cough, pneumonia, and syphilis, as detailed in this work. This imaging modality enables the visualization of alterations in the cortical layer of the bone, facilitating a crucial evaluation of disease progression. 

In the case of X-ray imaging of the humerus, it was observed that in a patient suffering from syphilis, there was evidence of slight widening, contour irregularities, and lytic changes in the central part of the distal areas [48]. This destructive lesion was observed in the lateral aspect of the metaphysis, where a calcified linear periosteal reaction was also present. Conversely, no radiopaque bands were identified in the metaphysis [82]. 

However, an X-ray examination of the shaft of the left ulna revealed the presence of a destructive lesion accompanied by an exuberant lamellar periosteal reaction, described as ‘onion skin’ [82]. 

Additionally, syphilitic lesions may be identified during the diagnosis of long bones, including the femur and tibia. The photographs illustrate a periosteal reaction that is generalized in nature, as well as zones of extensive obliteration of the medullary cavity that are sclerotic. Additionally, the bones exhibited thickened periosteum and narrowed, partially obliterated medullary cavities [19]. In some cases, the outer layers of the femoral cortical bone become detached in their distal region [48]. 

A radiological examination revealed the presence of a homogeneous spongy network in the hemifrontal region, although some irregular areas were also discernible. The periosteal junction area of the vault on the right hemifrontal and both parietals exhibited extensive new bone formation [6]. Upon examination of the parietal bones, star-shaped, clear areas with surrounding sclerotic bone were observed, which exhibited an identical appearance to the healed lesions in modern cases [53]. 

#### 3.4.2. CT Imaging

Computed tomography (CT) is a method that provides superior imaging of pathological alterations in the bones in comparison to a conventional X-ray. In the analysis and imaging of bone samples, scanning is conducted in layers. This is employed to reinforce the case for infectious systemic diseases and to visualize alterations in the cortical layer of the bone [98]. 

In cases of Treponema pallidum infection, the skull CT imaging may reveal evidence of osteolysis and lesions that penetrate the frontal bone, resulting in the formation of serpiginous cavities. Such lesions may subsequently progress to the right orbit, where remodeling of the nasal aperture is evident, as well as the inferior nasal conchae. On CT scans, the lesions are shown as translucencies and thinning in the cortical and diploë bone. The CT scan even revealed a process of already ongoing inflammation within the anterior cranial fossa and the left side of the middle cranial fossa [24]. 

In some cases, indicating a more advanced stage of progression, on the frontal bone, the destructive changes are significantly more severe, exhibiting clustered pitting, erosion of the bone, and marked cavitation. Additionally, nodule formation may occur in some more discrete areas. The aforementioned lesions and their characteristic locations are consistent with those observed in cases of venereal syphilis. This form of syphilis primarily affects the frontal and parietal bones of the neurocranium, where the periosteum adheres closely to the bone surface [18]. 

It should be noted, however, that the spirochetes do not merely penetrate the aforementioned skull bones; they also invade the occipital bone where a ‘worm-eaten’ appearance is produced [3,70]. 

Additionally, alterations resulting from syphilis can be observed in the femurs, as evidenced by computed tomography (CT) imaging, which reveals focal destruction and a cavity. Upon sagittal sectioning, the femurs display extensive cavitations that penetrate deeply into the cortex and regions where the cortex has thickened, resulting in a reduction in the marrow cavity area. Also, in the case of femurs, spirochetes can cause a ‘worm-like’ appearance of the bone with rounded edges [3]. 

Imaging studies of other bones, for example, the tibia, reveal osteolytic changes in the new bone in the anterior part and a reduction in the density of the original bone. In contrast, a CT scan of the ischium revealed the presence of diffuse lytic structures, initially visible as pitting concentrated on the bone surface [72]. 

#### 3.4.3. Micro-CT Imaging

Micro-CT (ICT) represents a novel three-dimensional imaging method that facilitates high-resolution imaging of samples. Although this new solution is expensive, it offers a wider range of imaging possibilities due to the use of a focused X-ray beam and a surface detector. This enables the visualization and analysis of defects in bone microarchitecture caused by Treponema pallidum. It can thus be concluded that the method is primarily employed for investigations into the altered mechanical properties of bone [99]. 

It has been demonstrated in studies that tertiary syphilis is responsible for at least one cranial perforation, which can be identified by comparing macerated skulls with those that are not affected. The majority of these perforations affect the visceral skull, as evidenced by the observation that they are present in up to 80% of the 20 skulls that were analyzed. Conversely, the neurocranium was similarly affected by syphilitic infection in the same study, with 50% of individuals exhibiting evidence of this. Another notable characteristic of the infection is the complete porosity of the bones observed in patients. Additionally, thinning of the cerebral cortex, sclerotic reorganization, and loss of cortex can be observed in the majority of individuals [13]. The use of 3D μ-CT images has revealed a common phenomenon, namely, osteolytic destruction of the outer lamina and the formation of cavities within the diploe. Subsequently, the entire cavity may become filled with secondary woven bone. As a consequence of these processes, the outer and inner lamina may undergo thickening (total diameter 10 mm instead of approximately 4 mm), and the area may exhibit a relatively smooth surface. The thickening is visible in the bulging part of the outer lamina and the thinner part of the inner lamina. ICT provides information on the thickening of the beads and their isotropic arrangement. Furthermore, the center of the dipole displays additional indentations of varying dimensions [68]. The non-destructive imaging method has proved invaluable in identifying a wide range of skeletal disorders caused by tertiary syphilis [13]. 

Nevertheless, this imaging technique also permits the observation of lesions resulting from congenital syphilis. All specimens exhibited the presence of extensive new bone on both the periosteal and intraperitoneal surfaces. Conversely, the separation of growth plates was observed in some long bones [22]. 

In conclusion, micro-CT imaging offers significant potential for the diagnosis of bone diseases, enabling the performance of a range of studies without the destruction of specimens. This method can make a substantial contribution to the advancement and expansion of knowledge in this field. 

### 3.5. Genetic Techniques

Nine of the included papers concerned the detection of spirochete DNA in fossil material. In all of them, an attempt was made to extract ancient DNA (aDNA) and then multiply it by PCR and detect it. Molecular studies were always accompanied by macroscopic methods and, in some publications, also by radiological methods. The remains came from areas of the US, UK, France, Spain, Hungary, Brazil, Mexico, and Canada, and their ages ranged between the 16th and first half of the 20th century. Skeletons of fetuses, newborns, children, adults, and the elderly were examined. A total of 153 individuals’ bones were examined who were suspected of having syphilis based on macroscopic changes [7,15,17,32,41,44,52,54,71,85]. 

The extraction of aDNA in the different studies followed a similar pattern. The sources of genetic material were bone fragments from the skull, teeth, ribs, femurs, and fibula. In one study, an unsuccessful attempt was made to extract Treponema pallidum aDNA from dental calculus [15]. It is worth noting the attempt to use LPA to purify genetic material in one study [41]. 

Detection of aDNA of Treponema pallidum was successful in five studies for a total of nine deceased [7,32,52,54,71]. These included one extremely fragmented skeleton, whose sex or age of death was not estimated by the authors, two fetal skeletons, four neonates, and two women whose ages were estimated to be <17 and <18 years. The common feature of the positive cases was a young skeletal age. However, in the same study, other samples subjected to the same procedure and of similar age yielded negative results. 

The search for T. pallidum aDNA in historical remains has advantages and limitations. The selection of specific primers (Eco47III) for PCR allows for an unambiguous diagnosis, which is challenging with macroscopic techniques [54]. For the detection of spirochete genetic material, 50 mg of bone dust is sufficient, while a larger fragment is needed for macroscopic evaluation. However, genetic techniques involve the permanent destruction of the collected sample [85]. Although the methods discussed are successful in searching for aDNA of some bacteria, e.g., M. tuberculosis [41], they do not show high efficacy for syphilis and are not part of routine testing. Researchers point out that the reason is a course of syphilis and the biochemical characteristics of the spirochete itself [41]. 

Treponema pallidum has a sensitive outer membrane [100], in contrast to most Gram-negative bacteria lacking lipopolysaccharide (LPS) [101]. This makes spirochetes susceptible to destruction by detergents, heat, and desiccation [102,103]. Barnes and Thomas (2001) attribute failure in their investigation to weak cell walls [17]. The researchers emphasize that the detection of the aDNA of T. pallidum is due to the focus on those who died of congenital syphilis, as younger patients have significantly more spirochetes in their skeleton [7]. The success of aDNA Treponema detection against the adopted criteria is based on epidemiological data, mostly the pathophysiology of the disease. Additional factors affecting the detection of aDNA are temperature and soil type [100,101]. 

### 3.6. Detection of Heavy Metals

Four of the included articles involved measuring mercury concentration in bone material and arsenic and lead in one case. In all cases, this method occurred together with macroscopic and radiological techniques. Bones in three of the research studies originate from cemeteries in Poland, and one concerns an anatomical collection from the USA. In total, measurements were carried out on one child, nine male, and eleven female skeletons, dated between the 15th century and the early 20th century. ICP-OES, LA-ICP-MS, and pXRF were used. In only one case was an arsenic level detected that was higher than in the remains without signs of syphilis, and in two cases, an elevated level of mercury was found [24,42]. The first of them is also the only case of acquired syphilis among those investigated; in the others, a congenital form of the disease was suspected. Mercury was used in the treatment of congenital and acquired syphilis, but because of its toxicity and unclear effectiveness, it was abandoned at the beginning of the 20th century. That is why measuring its concentration may seem an interesting diagnostic technique. 

### 3.7. Case Reports

A total of 31 case reports were analyzed in this paper [6,20,21,23,24,26,27,28,29,30,34,37,39,43,44,46,47,49,50,56,59,60,61,69,74,76,77,78,81,82,90]. The works analyzed in this subsection are original works, the subject of which was the examination of human remains for syphilitic lesions. The authors of the analyzed works used various diagnostic methods for this purpose, depending on their needs and capabilities. The applications and significant achievements of each method are approximated below.

The methods used in the reports consisted of macroscopic analysis of bone lesions, X-rays, CT scanning, microscopic or histological analysis of bone lesions, micro-CT scanning, mass spectroscopy, chemical analysis of the bone composition, immunohistochemical methods, aDNA detection, SEM scanning, and cadaver preparation. Diagnostic methods were identified and assigned to each case report. The methods used were counted for each study, the number of which is shown in Figure 5. The number of uses of each diagnostic method was then calculated and is shown in Figure 6. 

Among the case reports reviewed, macroscopic analysis of skeletal changes was also, as in the research articles reviewed, the primary method to begin the diagnostic process. In some unequivocal cases of finding pathognomonic lesions, it happened to be the only method needed to diagnose treponemal disease. In numerous of the cases reviewed, the authors diagnosed [2,3,5,13,25,48,63,72] or presumed [45,54,60] congenital syphilis. Hutchinson’s incisor, mulberry molar, darkened tooth enamel, and hypoplastic defects found in one specimen provided convincing confirmation of the diagnosis of congenital syphilis [2]. In some cases, radiological examinations were carried out to complement the macroscopic evaluation of bone lesions. 

With the help of regular X-ray imaging were observed changes characterizing chronic inflammatory processes, such as signs of osteoarthritis or Harris lines on tibiae and femora [63]. However, these are considered non-specific. Such changes are indicators of inflammation, which occur in many osteous disorders [62]. In addition to non-specific changes, it was possible to visualize Wimberger’s sign, which is typical of the congenital form of syphilis [25,63]. Micro-CT imaging was used to detect lesions of the still-hidden permanent dentition in a sub-adult individual [45]. CT scanning was the third most frequently chosen diagnostic method. Its use allowed confirmation of initial diagnoses made using macroscopic bone analysis [71] and also allowed the depth of the lesion to be determined [84]. SEM imaging was used to better visualize small lesions on the teeth [5]. Indirect immunofluorescence with human anti-treponema pallidum antibodies revealed syphilis in a Renaissance mummy [86]. The preparation of the cadaver was necessary to expose the subsequently analyzed bone lesions hidden under the mummified tissues in another mummy [13]. DNA analysis helped confirm the diagnosis of syphilis and even pinpoint the pathogen subspecies responsible—*Treponema pallidum* subsp. Pallidum [85].

## 4. Discussion

Macroscopic analysis is frequently employed as an initial sorting technique, categorizing skeletal remains by the presence of bone lesions characteristic of treponemal infection before detailed investigation using microscopic, radiological, histological, and molecular methods [1,3,5,13,15,16,24,25,41,45,54,60,71,84,85,86]. This ‘top-down’ methodology allows for an overarching perspective on the specimens under examination and the confirmation of macroscopic observations [24,82,86]. Researchers usually distinguish lesions associated with syphilis based on classical literature published in the 19th and 20th centuries authored by pioneering figures in the diagnosis of the disease, such as Hutchinson, Moon, Fournier, Williams, and Hackett [5,8,18,36,40,45,60,72,92,104]. 

Conducting a valid diagnosis requires considerable deduction and logical reasoning from the researcher, particularly since skeletal remains are often incomplete. Consequently, the best results are obtained by integrating macroscopic analysis of the lesions with the epidemiological, historical, and geographical context of the examined population [65]. Despite this innovative approach of differentiating between the treponematoses through the combination of macroscopic examination and population analysis [65], which is highly respected by some authors [58], other researchers still argue that different diagnostic methods may offer greater reliability [9,11,53]. On the contrary, it is essential to recognize that macroscopic analysis allows the examination of rare or highly damaged specimens, which cannot be sectioned and used for microscopic or molecular analysis [35,85,86]. Moreover, in certain cases, it remains the only diagnostic method available due to the absence of necessary equipment [80]. 

Analysis of the histopathological images allows a more comprehensive view of the researched diseases and confirms the diagnosis or negates it, leading to other possible etiologies. Further expansion of knowledge in the field of microscopic analysis in paleopathology can greatly benefit future research. However, useful newer analyses have shown that some of the “indicative” microscopic signs of syphilis, like Grenzstreifen and Polsters, are present in other diseases such as leprosy or nonspecific inflammatory diseases (e.g., hematogenous osteomyelitis). The difference relies on the manner of production of bone mass-syphilitic structures that are laid down gradually, solidly organized, as opposed to the rapid, disorganized structures in nonspecific inflammatory diseases [45]. It is worth noting that the differentiation of syphilis from other treponematoses, yaws, and endemic syphilis can also be difficult based only on those histological signs [24,45,65,86]. 

The use of computed tomography (CT) scanning as a diagnostic tool for syphilis, particularly at different stages of the disease, serves as a valuable complement to other diagnostic methods. Meanwhile, genetic techniques show promise, especially in diagnosing congenital syphilis and cases involving individuals who died at a young age. Ongoing research in this area may lead to significant advancements. However, it is important to recognize the limitations associated with these approaches. 

Syphilis, also known as “Great Imitator”, can be misdiagnosed as either a tumor, tuberculosis, bone inflammation, or leprosy, which could have been treated with arsenic compounds in the past. Arsenic was a far-ranging drug, too, because of its antiseptic, antipyretic, cholagogic, diastolic, calming, and tonic properties. Moreover, it was used as a component of paint or stained glass, as well as for leather and wood preservation [102]. It blocks the protein’s sulfhydryl groups, which leads to cell cycle arrest in the S phase and, therefore, to inhibition of bone regeneration [24,68,82,86]. Regarding mercury, it must be noted that in the Middle Ages and Early Modern period, mercury was also used as a laxative as well as a drug for conjunctivitis, corneal irritation, psoriasis, eczema, tinea, skin lesions, and others [103]. Therefore, researchers should take this into consideration while measuring heavy metal concentrations in bone material. 

## 5. Conclusions

The outcome of our review is a hierarchy of diagnostic procedures for identifying syphilis in skeletal remains. Macroscopic analysis, as it does not require expensive equipment, is frequently employed as a preliminary diagnostic tool used to identify remains with lesions characteristic of syphilis, positioning it at the top of the hierarchy. Microscopic examination of bone lesions plays a crucial role in distinguishing syphilitic symptoms from those of other diseases, thereby confirming or refuting the findings of macroscopic analysis. Radiological imaging complements macroscopic analysis effectively; however, with few exceptions, it identifies lesions that are sensitive but insufficiently specific to syphilis. In such cases, micro-CT offers substantial diagnostic potential, enabling the creation of 3D models of specimens without damaging them, which allows for detailed visualization and analysis of bone microarchitecture defects. The effective detection of T. pallidum aDNA is most feasible in remains from individuals who died of congenital syphilis. Ongoing research in this field holds significant promise for future advancements; however, the high cost of this method remains a limiting factor. Heavy metal detection, while non-specific, can serve as a supplementary method to other diagnostic approaches.

Differential diagnosis is based on differences in bone structure, which can sometimes be observed with the naked eye. However, the most reliable results are provided by microscopy. Syphilis does not produce pathognomonic symptoms with the exception of Hutchinson’s teeth, which are a sure indicator of congenital syphilis. Other important differences between the congenital and the acquired form of the disease are epiphyseal and metaphyseal erosion combined with cortical detachment and paradiaphyseal calcification, as well as curvature of the tibiae. Also, treponemal aDNA is usually obtained in cases of congenital syphilis.

This methodology provides a multifaceted approach to examining skeletal remains, allowing initial macroscopic observations to be verified and refined through more specific diagnostic techniques.

## Figures and Tables

**Figure 1 diagnostics-15-01116-f001:**
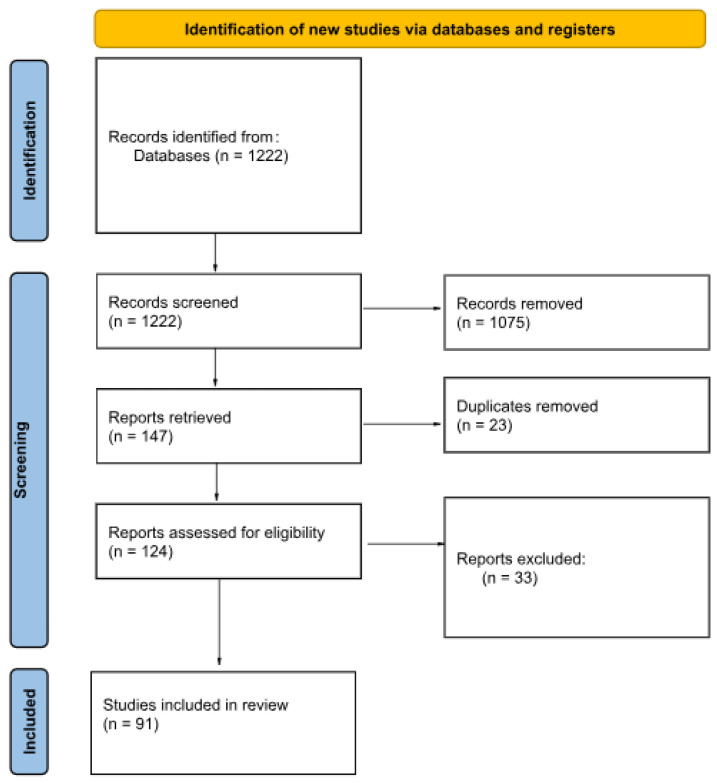
The procedure for including and excluding articles in review.

**Figure 2 diagnostics-15-01116-f002:**
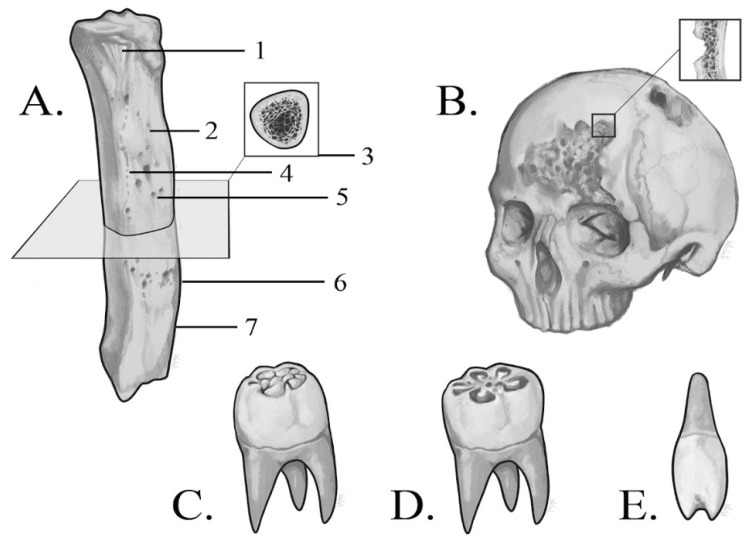
(**A**) 1. Irregular surface of bone due to remodeling of lesions. 2, 4, 5. Porosity of syphilitic bone. 3. Formation of lamellar bone in the medullar cavity. 6, 7. Pathological sclerosis of bone on its borders. (**B**) Caries sicca of the frontal bone. (**C**) Dome-shaped molar with a narrow occlusal surface and crowded cusps clustered at the center (**D**). Molar with penetrating sinuses and atrophied cusps. (**E**) Hutchinson’s incisor.

**Figure 3 diagnostics-15-01116-f003:**
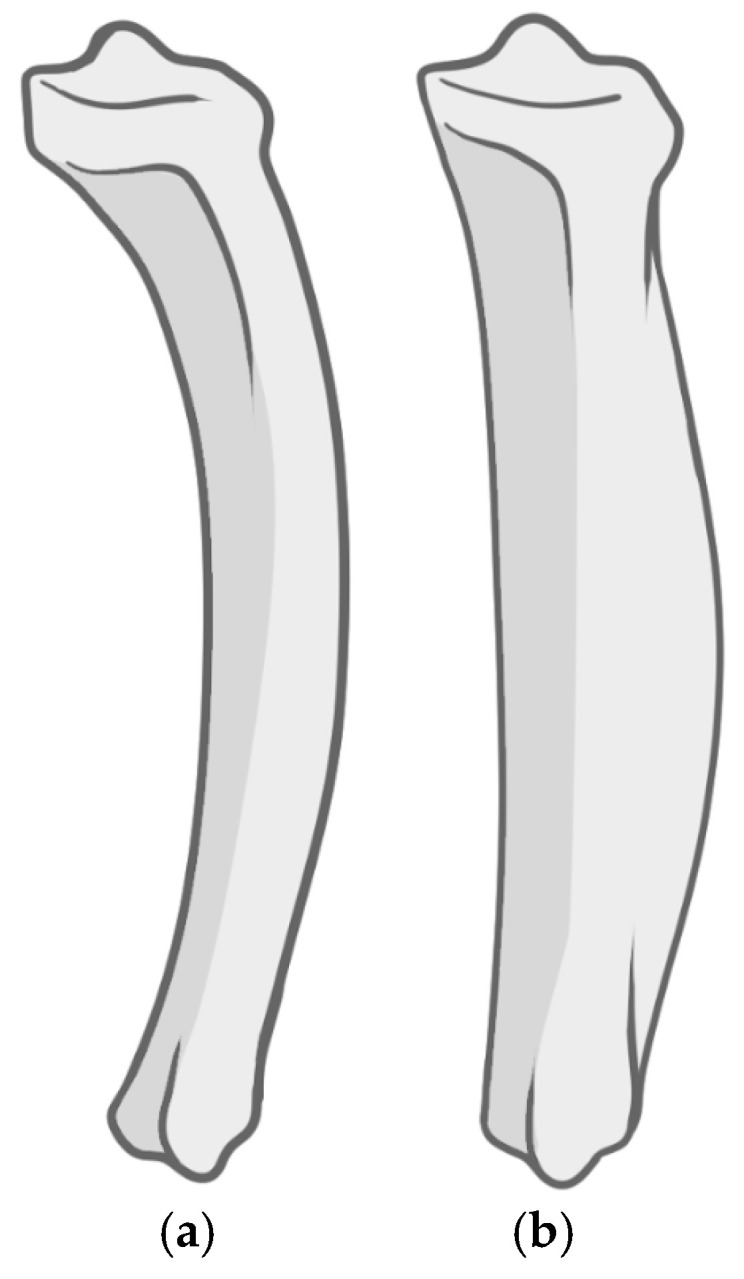
Syphilitic shin deformity in immature (**a**) and mature bone (**b**).

**Figure 4 diagnostics-15-01116-f004:**
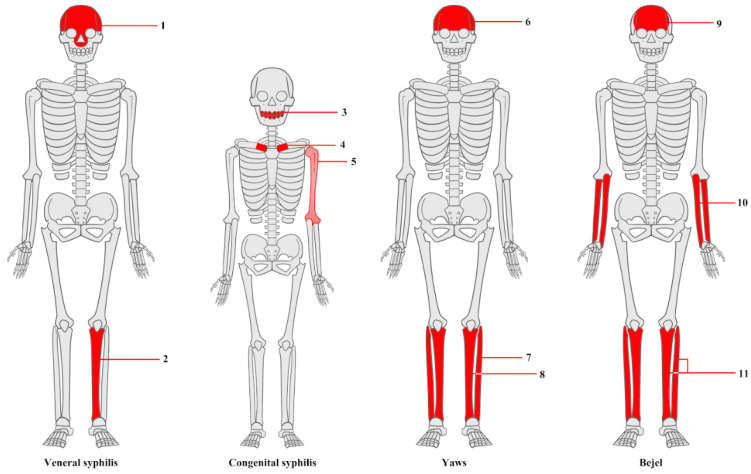
Differences in the localization of skeletal lesions in various treponematoses described in articles concerning the differentiation of treponemal diseases. Total of altered bones in case of each disease marked.

**Figure 5 diagnostics-15-01116-f005:**
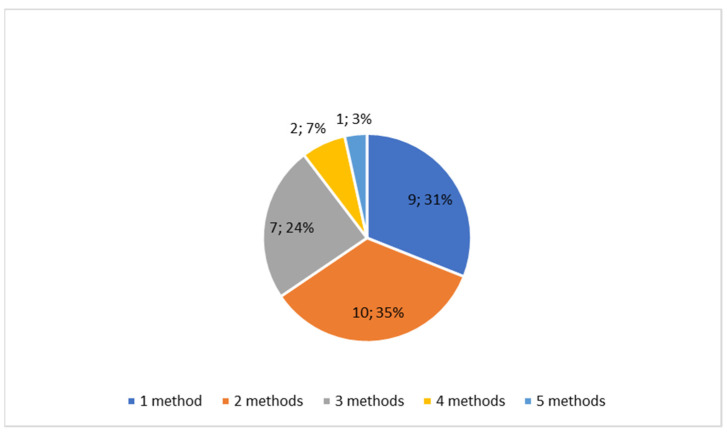
A diagram presenting the number of studies where various numbers of diagnostic methods were used. Two methods were usually a combination of macroscopic analysis and radiology. Three or more methods concerned macroscopic, radiological, and various other techniques.

**Figure 6 diagnostics-15-01116-f006:**
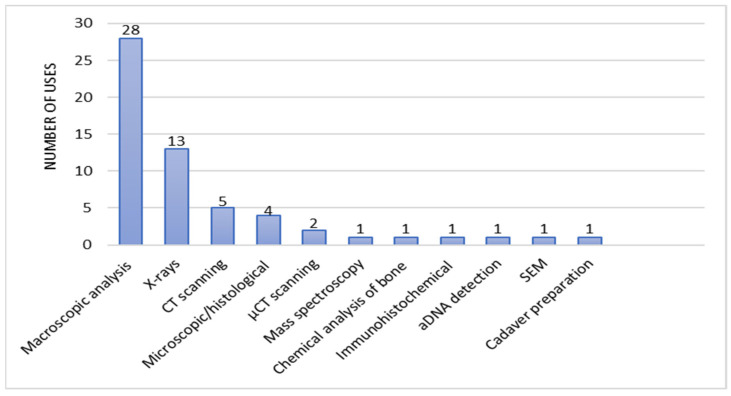
The number of uses of each diagnostic method.

**Table 1 diagnostics-15-01116-t001:** Characteristics of included studies. CE—Common Era, BCE—Before Common Era.

Article	Used Diagnostic Techniques	Age of Samples	Examined Materials
Anteric et al., 2014 [4]	Macroscopic analysis, X-ray imaging	Prehistory–19th century	Nearly complete skeletons
Assis et al., 2015 [16]	Macroscopic analysis, X-ray imaging	18th century	Cranium, upper limb bones, lower limb bones
Austin et al., 2024 [15]	Macroscopic analysis, PCR	19th–20th century	Dental calculus
Barnes and Thomas, 2006 [17]	Macroscopic analysis, PCR	19th–20th century	Cranium, femur, tibia, fibula, rib, sternum, vertebral column, pelvis, clavicle, tooth, osseus gumma
Biehler-Gomez et al., 2022 [18]	Macroscopic analysis, CT imaging	17th century	Crania
Buckley, 2000 [5]	Macroscopic analysis, X-ray imaging	800 CE	Cranium, clavicle, humerus, ulna, radius, bones of hands, femur, tibia, fibula, bones of feet
Buzhilova, 1999 [19]	Macroscopic analysis, X-ray imaging	16th century	Frontal bone, arm bones, forearm bones, tibia
Castro et al., 2020 [20]	Macroscopic analysis, X-ray imaging	190 ± 20 CE	Cranium, mandible, tooth, vertebra, clavicle, scapula, sternum, femur, tibia,
Castro et al., 2016 [21]	Macroscopic analysis	210 BCE	Sternum, vertebrae
Cole et al., 1955 [1]	Macroscopic analysis, X-ray imaging	ca. 600 CE–1300 CE	Cranium, femur, tibia
Cole et al., 2020 [22]	X-ray imaging, CT imaging, micro-CT imaging	19th century	Tibia, humerus, femur, ulna, fibula radius, rib, clavicle, ilium, mandible sacrum, scapula
Cole and Waldron, 2011 [23]	Macroscopic analysis, X-ray imaging	ca. 5th–7th century	Cranium, tibia, humerus, femur, ulna, fibula, radius, rib, clavicle, ilium, mandible, scapula, bones of feet
Dabernat et al., 2013 [2]	Macroscopic analysis, X-ray imaging	17th–20th century	Cranium, tibia, humerus, femur, ulna, fibula, radius, rib, clavicle, ilium, mandible, scapula, bones of feet
Dąbrowski et al., 2019 [24]	Macroscopic analysis, Microscopic analysis, CT imaging	16th–18th century	Cranium
El Najjar, 1979 [25]	Macroscopic analysis	100 CE–700 CE	Ribs, sternum, clavicle, scapula, shoulder, hip, sacroiliac joint, femur, knee, elbow
Erdal, 2006 [26]	Macroscopic analysis, X-ray imaging	13th century	Nearly complete skeleton
Fornaciari, 1999 [27]	Macroscopic analysis, microscopic analysis, PCR	16th century	Complete mummy
Fornaciari et al., 2020 [28]	Macroscopic analysis, CT imaging	16th century	Nearly complete skeleton
Fraberger et al., 2021 [13]	Macroscopic analysis, micro-CT imaging	Before 1909 CE	Crania
Frangos et al., 2011 [11]	Macroscopic analysis	16th–19th century	Clavicles
Gaul and Grossschmidt, 2014 [29]	Macroscopic analysis	1765–1790 CE	Cranium, dentition, calcanei
Gaul et al., 2015 [30]	Macroscopic analysis	13th–14th century	Nearly complete skeleton
Gerszten et al., 1998 [31]	Macroscopic analysis	ca. 300 CE	Cranium
Giffin et al., 2020 [32]	Molecular techniques	15th–16th century	Teeth
Guedes et al., 2018 [7]	Macroscopic analysis, molecular techniques	18th–19th century	Crania, mandibles
Hacket, 1975 [33]	Macroscopic analysis, microscopic analysis	Unspecified	Crania, long bones
Henkel et al., 2020 [34]	Macroscopic analysis, molecular techniques	19th–early 20th century	Complete skeleton
Hernandez and Hudson, 2015 [3]	Macroscopic analysis, X-ray imaging, CT imaging	17th–19th century	Crania, mandibles, humeri, radii, ulnae, femora, tibiae
Hillson et al., 1998 [35]	Macroscopic analysis, microscopic analysis	Unspecified	Teeth
Ioannou et al., 2017 [36]	Macroscopic analysis, X-ray imaging, molecular techniques	Early 20th century	Teeth
Ioannou et al., 2015 [37]	Macroscopic analysis	19th century	Teeth, cranium, mandible, clavicle, ribs, vertebrae
Ioannou and Henneberg, 2017 [38]	Macroscopic analysis	1912 CE–1928 CE	Complete skeletons
Ioannou and Henneberg, 2016 [39]	Macroscopic analysis, X-ray imaging	Early 20th century	Teeth, mandible, tibia, fibula, humeri, radius, ulnae, Femora, ilium
Ioannou et al., 2018 [12]	Macroscopic analysis	100 CE–250 CE, 1390 CE–1440 CE, 8th–2nd century BCE, 13th century	Teeth
Jacobi et al., 1992 [40]	Macroscopic analysis, SEM imaging	ca. 1660 CE–1820 CE	Permanent incisors, first molars
Jäger et al., 2022 [41]	Macroscopic analysis, DNA sequencing	1755 CE	Ribs, cranium
Kepa et al., 2012 [42]	Macroscopic analysis, LA–ICP–MS, Spectrometry	14th–19th century	Teeth, temporal diaphysis, phalanx, sphenoid bone, zygomatic arch, squama temporalis, femur, humeral bone, rib
Klaus and Ortner, 2014 [43]	Macroscopic analysis, Magnified macroscopic analysis	ca. 1535 CE	Cranial vault, vertebral column, os coxae, scapula, clavicle, ribs, sternum, humerus, radius, ulna, hands, femur, tibia, fibula, feet
Kolman, et al., 1999 [44]	Macroscopic analysis, ELISA, DNA sequencing	1759 ± 50 CE	Tibia, femur
Lewis, 1994 [45]	Macroscopic analysis, X-ray imaging, histologic examination	500 BC–300 CE	Nearly complete skeletons
Lopez, et al., 2017 [46]	Macroscopic analysis, Microscopic analysis, CT scan,	879 CE–1001 CE	Cranium, mandible, teeth, humerus, scapulae, clavicle, ribs, sacrum, os coxae
Lopes et al., 2010 [47]	Macroscopic analysis	19th century	Cranium
Malgosa, et al., 1996 [48]	Macroscopic analysis, Microscopic analysis, X-ray imaging	1550 CE–1900 CE	Hemifrontal, humerus, femur
Mansilia and Pijoan, 1995 [49]	Macroscopic analysis, X-ray imaging	17th–18th century	Cranium, mandible, molars, tibia, fibula, femur, humerus, radius, ulna
Marden and Ortner, 2011 [50]	Macroscopic analysis	950 CE–1150 CE	Cranium, mandible, clavicles, scapulae, vertebrae, ribs, manubrium, sacrum, os coxae, fifth metacarpal, phalanges, radius, ulna, third metacarpal, fourth metacarpal, tibiae, fibulae, tali, calcanei, naviculars, femur, distal right femur, cuboids, first cuneiform, second cuneiform, third cuneiform, metatarsals, pedal phalanges, patella
Mays, et al., 2003 [51]	Macroscopic analysis, X-ray imaging	1295 CE–1445 CE and 1445 CE–1520 CE	Tibiae, frontal bone, teeth, clavicles, ribs, ulna, thoracic vertebrae, Radius, carpals, femur, fibula, Cranium
Meffray et al., 2019 [52]	Macroscopic analysis, molecular analysis	1837 CE–1867 CE	Cranium, scapulae, ilium bones, long bones of the limbs
Mitchell, 2003 [53]	Macroscopic analysis, X-ray imaging	1290 CE–1420 CE	Parietal bones, occipital bone fragment
Montiel, et al., 2012 [54]	Macroscopic analysis, Molecular Analysis, PCR, DNA sequencing, Radiological examination	16th–17th century	Femur, frontal bone, humerus
Nystrom, 2011 [55]	Macroscopic analysis	early to mid-19th century	Incisors, canines, and first permanent molars, tibiae
Palfi et al., 1992 [6]	Macroscopic analysis, X-ray imaging	3th–5th century	Fetal skeleton
Patel and Mitchell, 2007 [56]	Macroscopic analysis	1886 CE	Cranium
Pietrobelli et al., 2020 [57]	Macroscopic analysis, CT imaging, micro-CT imaging	late-14th to the mid-16th century	Crania, humeri, tibiae, fibulae, radii, ulnae, and femora
Pineda et al., 2009 [58]	Macroscopic analysis, X-ray imaging	1100 CE–1300 CE	Crania, tibiae, femora, and fibulae
Pineda et al., 1998 [59]	X-ray imaging, CT imaging	1000 CE–1600 CE	Mummified body, cranium
Radu and Soficaru, 2016 [60]	Macroscopic analysis, microscopic analysis, SEM imaging	Early 16th to first half of 19th century	First deciduous molars, permanent incisors, and canines
Radu et al., 2015 [8]	Macroscopic analysis	Early 16th to first half of 19th century	Crania, clavicles, ribs, vertebral bodies, femora, humeri, tibiae, fibulae, radii, ulnae, phalanges, metacarpals, ribs, pelvic bones, calcanei
Rissech et al., 2013 [61]	Macroscopic analysis X-ray imaging CT imaging	2nd to 3rd century	Nearly complete skeleton
Rothschild and Heathcote, 1993 [62]	Macroscopic analysis X-ray imaging	1500 CE	Tibiae, fibulae, femora, radii, ulnae, humeri, metacarpals, metatarsals, phalanges, Crania, clavicles
Rothschild and Rothschild, 1995 [63]	Macroscopic analysis	1795 CE–1945 CE	Femora, tibiae, fibulae, humeri, radii, ulnae, clavicles, ribs, hand bones, foot bones, crania
Rothschild and Jellema, 2020 [64]	Macroscopic analysis, microscopic analysis	20th century	Tibiae
Rothschild et al., 2000 [65]	Macroscopic analysis, magnified macroscopic analysis	2650 BCE–1400 CE	Tibiae, femora, fibulae, humeri, radii, ulnae, hand bones, foot bones, clavicles
Rothschild and Rothschild, 1997 [66]	Macroscopic analysis	1350 CE–1450 CE	Tibiae, fibulae, femora, hand bones, foot bones
Rothschild et al., 2011 [67]	Macroscopic analysis	8000 BCE–1200 CE	Tibiae, hand, and foot bones
Rühli et al., 2007 [68]	Macroscopic analysis, microscopic analysis, X-ray imaging, micro-CT imaging	Early 20th century	Crania
Salesse et al., 2019 [69]	Isotope analysis	19th century	Tibiae, femora
Sarhan et al., 2023 [70]	CT imaging, histological analysis, molecular analysis	1787 CE	Mummified body
Schuenemann et al., 2018 [71]	Macroscopic analysis, PCR, aDNA extraction	After 1650 CE	Long bones
Schwarz et al., 2013 [72]	Macroscopic analysis, CT imaging	1050 CE to 1530 CE	Nearly complete skeletons
Shuler, 2011 [73]	Macroscopic analysis, magnified macroscopic analysis	1796–1801 CE and 1811–1825 CE	Nearly complete skeletons
Šlaus and Novak, 2007 [74]	Macroscopic analysis	1478–1636 CE	Nearly complete skeleton
Somers et al., 2017 [75]	Macroscopic analysis	14th to 16th century and 17th to 19th century	Nearly complete skeletons
Souza, et al., 2006 [76]	Macroscopic analysis, X-ray imaging,	18th century	Complete mummy
Stirland, 1991 [77]	Macroscopic analysis, X-ray imaging	1100s to 1468 CE	Nearly complete skeleton
Steyn and Henneberg, 1995 [78]	Macroscopic analysis	1000–1300 CE	Nearly complete skeleton
Suzuki, 1984 [79]	Macroscopic analysis, X-ray imaging	Latter half of 16th century	Cranium, right femur, left tibia and fibula, scapula, rib, vertebrae, hip bone, fibula, and other unidentified fragments
Suzuki et al., 2005 [80]	Macroscopic analysis	Bronze Age	Crania, humeri, ulnae, radii, femora, tibiae, and fibulae
Szczepanek, et al., 2019 [81]	Macroscopic analysis	17th–18th century	Nearly complete skeletons
Tomczyk, et al., 2015 [82]	Macroscopic analysis, Microscopic analysis, X-ray imaging, LA–ICP–MS laser ablation,	1790–1812 CE	Teeth, Cranium, Mandible, Clavicles, Ribs, Cervicothoracic spine, scapulae, humeri, ulna, phalanges, ilium
Vargová, et al., 2021 [83]	Macroscopic analysis, X-ray imaging, Histological examination,	13th–19th century	Pelvic bone, lumbar vertebrae, Cranium, teeth, Tibia, femur
Vargová, et al., 2014 [84]	Macroscopic analysis, X-ray analysis, CT scan,	16th–17th century	Tibia, Lumbar vertebrae, Cranium, Ilium, Proximal femur, Fibula, Radius, Ulna, Ribs, Teeth
Von Hunnius, et al., 2007 [85]	Macroscopic analysis, PCR, DNA sequencing,	1861–1865 CE, 1450–1475 CE, 1300–1450 CE, ca. 1850 CE, ca. 1450 CE	Tibia, fibula, rib, cranium, long bone, Cranium, humerus, teeth, scapula, radius, femur
Von Hunnius, et al., 2006 [86]	Macroscopic analysis, Microscopic analysis, Histological examination,	1300–1450 CE	Fibula, humerus, long bone, Tibia, thoracic vertebrae, cervical vertebrae, upper thoracic vertebrae, ulnae, radii, Clavicles, Femur, tibiae, fibulae, Cranium, Mandible
Walker, 1983 [87]	Macroscopic analysis, X-ray imaging	515 BCE	Manubrium, Clavicle, thoracic vertebrae
Walker, et al., 2015 [9]	Macroscopic analysis	ca. 1120–1539 CE	Nearly complete skeletons
Weston, 2009 [88]	Macroscopic analysis, microscopic analysis	19th century	Femora, tibiae, and fibulae
Weston, 2008 [89]	Macroscopic analysis, X-ray imaging	18th to 19th century	Femora, tibiae, and fibulae
Woo, et al., 2019 [90]	Macroscopic analysis, X-ray imaging	19th century	Cranium, mandible, all major limb bones, pectoral and pelvic girdles, and some vertebrae
Zhou, et al., 2022 [91]	Macroscopic analysis, magnified macroscopic analysis	618–1279 CE	Crania, long bones
Zuckerman, 2017 [92]	Macroscopic analysis	1666–1853 CE	Crania, long bones
Zuckerman, 2016 [93]	Macroscopic analysis,	1666–1853 CE	Left femora

## Data Availability

No research data are available, due to privacy.

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
