# Peer review of "Diagnostic Methods Used in Detecting Syphilis in Paleopathological Research—A Literature Review"

_diagnostics, 2025, doi:10.3390/diagnostics15091116_

Round 1

Reviewer 1 Report

Comments and Suggestions for Authors

The article aims to provide an overview of the latest methodology regarding syphilis research on skeletal material. The article provides information on signs of syphilis infection on skeletal tissue. Although the introduction orients the readers to the topic, more information is required. The gap in literature the manuscript seeks to close was not identified. The authors did not provide information on how big the problem of skeletal manifestations of syphilis is. The methodology used in the literature review is not clearly explained, and the search results presented in the figure and narrative are conflicting. The authors did not provide a conclusion to their manuscript. This article does not provide any originality since it states what is already known, raising questions about its significance to readers. It might be more significant if the authors critically analyze the findings in the different included articles.

MAJOR REVISIONS

INTRODUCTION

  1. To support the relevance of this manuscript, the authors must provide statistics on bone involvement among people infected with syphilis.
  2. The authors must provide information on the stages of syphilis that are associated with skeletal involvement.
  3. The authors must identify a gap in literature they hope to close.

MATERIALS AND METHODS

  1. The study design must be explicitly stated.
  2. The authors wanted to provide the latest methodology regarding syphilis research on skeletal material. Therefore, they should provide information on the period in which the articles for the literature review were published and justify this period.
  3. Numbers in Figure 1 are not similar to those in the narrative. For example, in the narrative, the authors state that 125 articles remained after removing duplicates. However, in the figure, the number is 124.
  4. The authors must provide details on how data were extracted and analyzed.

RESULTS

  1. Part of 3.6 discusses the characteristics of the included studies. A sub-section on the characteristics of the included studies must be provided at the beginning of this section.

CONCLUSION

  1. There should be a conclusion section at the end of the manuscript.

MINOR REVISIONS

  1. In lines 454-455, the authors stated that there were 9 deceased. However, they provide age information for only 8. They should provide the age information on the other deceased.
  2. In line 490, the authors state XX century. This should be changed to numbers.

Author Response

Response to Reviewer 1 Comments

Dear reviewer,

Thank you very much for time spent on reviewing this manuscript and valuable comments. We agree with all of them. Please find the detailed responses below and the corresponding corrections in the re-submitted file.

Major revisions

  1. Comment 1: To support the relevance of this manuscript, the authors must provide statistics on bone involvement among people infected with syphilis.
    • Response 1: We included this information in introduction. Line 56-61.
  2. Comment 2: The authors must provide information on the stages of syphilis that are associated with skeletal involvement.
    • Response 2: We included this information in introduction. Line 50-55.
  3. Comment 3: The authors must identify a gap in literature they hope to close.
    • Response 3: Our aim was to describe, compare and summarize techniques used in diagnosing syphilis in osseous remains. We included it in introduction. Line 75-80.
  4. Comment 4: The study design must be explicitly stated.
    • Response 4: We extended Materials and Methods section, so it describes explicitly process of collecting articles, data extraction and its’ analysis. Lines 82-111
  5. Comment 5: The authors wanted to provide the latest methodology regarding syphilis research on skeletal material. Therefore, they should provide information on the period in which the articles for the literature review were published and justify this period.
    • Response 5: We included this information in Materials and Methods section. Lines 82-86.
  6. Comment 6: Numbers in Figure 1 are not similar to those in the narrative. For example, in the narrative, the authors state that 125 articles remained after removing duplicates. However, in the figure, the number is 124.
    • Response 6: Thank you for pointing this out. It was typographic error, we corrected it. Line 107.
  7. Comment 7: The authors must provide details on how data were extracted and analyzed.
    • Response 7: We extended Materials and Methods section, so it describes explicitly process of collecting articles, data extraction and its’ analysis. Lines 82-111
  8. Comment 8: Part of 3.6 discusses the characteristics of the included studies. A sub-section on the characteristics of the included studies must be provided at the beginning of this section.
    • Response 8: We included this sub-section. Line 544-549
  9. Comment 9: There should be a conclusion section at the end of the manuscript.
    • Response 9: We included conclusion section, summarizing outcome of our review. Lines 655-681.

Minor revisions

  1. Comment 1: In lines 454-455, the authors stated that there were 9 deceased. However, they provide age information for only 8. They should provide the age information on the other deceased.
    • Response 1: We included information on the last deceased. Lines 504-506.
  2. Comment 2: In line 490, the authors state XX century. This should be changed to numbers.
    • Response 2: It was changed to numbers. Line 541.

We hope we stated every change clearly and managed to upgrade our manuscript following your comments. Thank you, for taking time for it.

Reviewer 2 Report

Comments and Suggestions for Authors

Dear editors and authors,

Thank you for the opportunity to read this engaging and well-researched review of the literature on syphilis. The authors present valuable and insightful information on this important topic, which plays a crucial role in anthropological and paleopathological contexts for a better understanding of the health status of historical populations. The work is well structured, clearly organized and well written. However, there are some issues that require further attention and minor revision to improve the overall quality of the study. Please, see my recommendations below. Overall, the article is worth publishing after a minor revision and incorporation of the suggested improvements.

I suggest that the authors change the key words slightly, e.g. anthropological methods, paleopathological analysis, periostitis, syphilis nodules, long bones...

  • In the introduction, it is important to mention (as you do in the abstract) which bones are most commonly affected. As far as I know, this is then mentioned in the results section but mention it briefly in the introduction as well. I also understand that the tertiary stage of syphilis is mainly found in the skeletal material, but it might be useful and would give an overall view of syphilis if the authors also briefly mention the stages (primary, secondary (congenital and venereal) syphilis). These are also mentioned in the results section, but please also mention them briefly in the introduction.
  • It might be useful to compare the prevalence of syphilis in different historical populations, if data are available. Are there studies from other European areas that could be compared?
  • In the Materials and methods section – were there specific inclusion criteria on date for the papers used in the present study?
  • I recommend that the authors specify which osteologic features are most reliable for the differential diagnosis of syphilis compared to other infectious diseases (e.g., leprosy or tuberculosis). Several diagnostic features are mentioned in the text, but their reliability is not always clearly justified.
  • Hutchinson teeth – more information, that these are present in the permanent dentition should be added.
  • Please indicate in Figure 5 what “1 method",” “2 methods” etc. mean, i.e. what combinations of methods were used.
  • Line 515 – You mention the Harris lines, but it would be helpful if you would specify on which bones these were observed. Also, please explain their significance for the analysis of syphilis. How do the Harris lines contribute to the differential diagnosis of syphilis and how are they interpreted in the context of this disease? Clarification of whether they are considered a direct or indirect indicator of syphilis-related physiologic stress would enrich the discussion.
  • The text could more clearly distinguish the manifestations of congenital and acquired syphilis in skeletal material. What are the main differences? This is partially mentioned in the text, but not systematically addressed.

Author Response

Response to Reviewer 3 Comments

Dear reviewer,

Thank you very much for time spent on reviewing this manuscript and valuable comments. We agree with them. Please find the detailed responses below and the corresponding corrections in the re-submitted file.

  1. Comment 1: I suggest that the authors change the key words slightly, e.g. anthropological methods, paleopathological analysis, periostitis, syphilis nodules, long bones...
    • Response 1: We changed the key words, including these you suggested. Lines 37-38.
  2. Comment 2: In the introduction, it is important to mention (as you do in the abstract) which bones are most commonly affected. As far as I know, this is then mentioned in the results section but mention it briefly in the introduction as well. I also understand that the tertiary stage of syphilis is mainly found in the skeletal material, but it might be useful and would give an overall view of syphilis if the authors also briefly mention the stages (primary, secondary (congenital and venereal) syphilis). These are also mentioned in the results section, but please also mention them briefly in the introduction.
    • Response 2: We extended introduction. Now it includes natural progression of syphilis and bone involvement. Lines 50-61.
  3. Comment 3: It might be useful to compare the prevalence of syphilis in different historical populations, if data are available. Are there studies from other European areas that could be compared?
    • Response 3: Our aim was to describe, compare and summarize techniques used in diagnosing syphilis in osseous remains. The methods described are applicable to the examination of remains from an extensive time span and from different locations around the world. Often the same or similar methods are used to diagnose very different cases. After much thought, we concluded that, because of this universality, information on incidence in different historical populations does not quite fit the context of the work.
  4. Comment 4: In the Materials and methods section – were there specific inclusion criteria on date for the papers used in the present study?
    • Response 4: We clarified it in Materials and Methods section. Lines 82-86.
  5. Comment 5: I recommend that the authors specify which osteologic features are most reliable for the differential diagnosis of syphilis compared to other infectious diseases (e.g., leprosy or tuberculosis). Several diagnostic features are mentioned in the text, but their reliability is not always clearly justified.
    • Response 5: We included conclusion section, which summarizes outcome of our review, including topic of differentiation. We hope this would make it clear. Lines 671-674.
  6. Comment 6: Hutchinson teeth – more information, that these are present in the permanent dentition should be added.
    • Response 6: Thank you, for pointing this out. We included this information. Lines 208-211.
  7. Comment 7: Please indicate in Figure 5 what “1 method",” “2 methods” etc. mean, i.e. what combinations of methods were used.
    • Response 7: We extended description of this figure. Line 565.
  8. Comment 8: Line 515 – You mention the Harris lines, but it would be helpful if you would specify on which bones these were observed. Also, please explain their significance for the analysis of syphilis. How do the Harris lines contribute to the differential diagnosis of syphilis and how are they interpreted in the context of this disease? Clarification of whether they are considered a direct or indirect indicator of syphilis-related physiologic stress would enrich the discussion.
    • Response 8: We clarified this topic. Lines 587-588.
  9. Comment 9: The text could more clearly distinguish the manifestations of congenital and acquired syphilis in skeletal material. What are the main differences? This is partially mentioned in the text, but not systematically addressed.
    • Response 9: We clarified it in conclusion. Lines 674-678.

We hope we stated every change clearly and managed to upgrade our manuscript following your comments. Thank you, for taking time for it.

Round 2

Reviewer 1 Report

Comments and Suggestions for Authors

Thank you for addressing my comments. However, the authors must still address the following issues.

  1. State what type of review this study is. Merely mentioning that it is a review is not enough.
  2.  The authors should explain what data was extracted from each article and how it was analyzed. Was it thematic or what?
  3. The authors must summarize all the included articles in table form, stating the authors, study design, and key findings in table form at the beginning of the results section. This should be under the 'characteristics of included studies' subheading.

Author Response

Response to Reviewer 1 Comments

Dear reviewer,

Thank you very much for your accurate comments. Please find detailed response, as well as changes in the manuscript.

  1. Comment 1: State what type of review this study is. Merely mentioning that it is a review is not enough.
    • Response 1: We included this information in Materials and Methods section. Line 82.
  2. Comment 2: The authors should explain what data was extracted from each article and how it was analyzed. Was it thematic or what?
    • Response 2: We included this information in Materials and Methods section. Line 112 – 120
  3. Comment 3: The authors must summarize all the included articles in table form, stating the authors, study design, and key findings in table form at the beginning of the results section. This should be under the 'characteristics of included studies' subheading.
    • Response 3: Thank you for this suggestion. We included such table at the beginning of the results section, under subheading you proposed. Line 125

We hope we stated every change clearly and managed to upgrade our manuscript following your comments. Thank you, for taking time for it.